# Robust Segmentation of Lung Proton and Hyperpolarized Gas MRI with Vision Transformers and CNNs: A Comparative Analysis of Performance Under Artificial Noise

**DOI:** 10.3390/bioengineering12080808

**Published:** 2025-07-28

**Authors:** Ramtin Babaeipour, Matthew S. Fox, Grace Parraga, Alexei Ouriadov

**Affiliations:** 1School of Biomedical Engineering, Faculty of Engineering, The University of Western Ontario, London, ON N6A 3K7, Canada; rbabaeip@uwo.ca (R.B.); gparraga@uwo.ca (G.P.); 2Department of Physics and Astronomy, The University of Western Ontario, London, ON N6A 3K7, Canada; mfox28@uwo.ca; 3Lawson Health Research Institute, London, ON N6C 2R5, Canada; 4Department of Medical Biophysics, The University of Western Ontario, London, ON N6A 3K7, Canada; 5Robarts Research Institute, London, ON N6A 5B7, Canada

**Keywords:** medical image segmentation, Vision Transformers, hyperpolarized gas MRI, CNN, SegFormer, image noise, lung imaging, deep learning, attention mechanism, robustness

## Abstract

Accurate segmentation in medical imaging is essential for disease diagnosis and monitoring, particularly in lung imaging using proton and hyperpolarized gas MRI. However, image degradation due to noise and artifacts—especially in hyperpolarized gas MRI, where scans are acquired during breath-holds—poses challenges for conventional segmentation algorithms. This study evaluates the robustness of deep learning segmentation models under varying Gaussian noise levels, comparing traditional convolutional neural networks (CNNs) with modern Vision Transformer (ViT)-based models. Using a dataset of proton and hyperpolarized gas MRI slices from 56 participants, we trained and tested Feature Pyramid Network (FPN) and U-Net architectures with both CNN (VGG16, VGG19, ResNet152) and ViT (MiT-B0, B3, B5) backbones. Results showed that ViT-based models, particularly those using the SegFormer backbone, consistently outperformed CNN-based counterparts across all metrics and noise levels. The performance gap was especially pronounced in high-noise conditions, where transformer models retained higher Dice scores and lower boundary errors. These findings highlight the potential of ViT-based architectures for deployment in clinically realistic, low-SNR environments such as hyperpolarized gas MRI, where segmentation reliability is critical.

## 1. Introduction

Medical image segmentation is a crucial technique in the field of medical imaging, where an image is divided into distinct regions or segments to identify and isolate specific structures or areas of interest. The goal of segmentation is to transform complex images into simpler, more meaningful representations that can be easily analyzed.

One advanced application of medical image segmentation is in hyperpolarized gas magnetic resonance imaging (MRI), particularly in lung imaging. Hyperpolarized gases like ^3^He (helium-3) and ^129^Xe (xenon-129) are used to produce highly detailed and high-contrast images of the lungs. Unlike traditional proton-based MRI, which struggles to capture clear images of the lungs due to the complex magnetic environment and rapid relaxation times of hydrogen atoms in the lung tissue, hyperpolarized gases offer a solution to these challenges. By polarizing the gas beyond its natural equilibrium magnetization, it becomes possible to visualize the airways and lung parenchyma with much greater clarity [1].

This imaging technique is especially beneficial for studying respiratory diseases like chronic obstructive pulmonary disease (COPD). COPD is characterized by airflow limitations caused by tissue damage and airway blockages, leading to irregular lung ventilation patterns [2]. Traditional imaging techniques, such as computed tomography (CT), have provided valuable insights into lung structure and pathology, helping to predict disease progression and treatment outcomes [3]. However, the reliance on ionizing radiation in CT raises concerns in long-term, repeated studies [4].

In contrast, hyperpolarized gas MRI offers a radiation-free alternative, allowing for repeated imaging without the associated risks [5]. Since its introduction in the 1990s, this technique has significantly improved our understanding of gas distribution in the lungs, especially in COPD patients [6]. By capturing the uneven distribution of inhaled gas, hyperpolarized gas MRI reveals regions of the lungs where airflow is restricted or obstructed. This can help identify functional and non-functional areas of the lungs, offering valuable information for diagnosing and managing COPD and other respiratory conditions.

The ability to segment these images and quantify regional abnormalities is critical for evaluating the effectiveness of treatments and tracking patient outcomes. Historically, manual segmentation techniques were used for this purpose, but they were time-consuming and prone to variability between observers. Modern advancements, including automated segmentation methods [7], have the potential to significantly enhance the precision and efficiency of this process, paving the way for better diagnosis and treatment of lung diseases.

Medical image segmentation plays a critical role in identifying anatomical structures and detecting diseases within the human body. However, one of the major challenges in this task is the presence of noise and artifacts that can significantly degrade the quality of the images, making segmentation difficult and reducing diagnostic accuracy [8]. The ability to segment medical images effectively is crucial because these images contain vital information about internal organs such as the heart, brain, and lungs, which can be essential for diagnosing life-threatening conditions. Inaccurate segmentation caused by noise can result in the loss of crucial information or the inclusion of irrelevant data, potentially leading to misdiagnosis or ineffective treatment.

Noise in medical images is particularly problematic because it introduces random changes in the pixel values, distorting the original features and lowering image contrast. Medical images, especially those generated by MRI, often have lower contrast compared to conventional images, making them more sensitive to noise [9]. When the structures being imaged are small or have low contrast, such as in the case of lung tissue, even minor distortions can severely affect segmentation accuracy. Handling noise in medical images is, therefore, a critical step to ensure that image quality remains high and that clinicians can make accurate assessments.

In the case of MRI, noise and artifacts can arise at various stages of image acquisition and processing. The process of acquiring an MRI scan involves capturing detailed images of the body’s internal structures, but due to factors such as equipment limitations, patient movement, and environmental interference, the images can be corrupted by different types of noise [10]. This can result in blurred edges, loss of detail, or artificial structures that make it difficult for segmentation algorithms to accurately distinguish between healthy and diseased tissues.

When focusing on hyperpolarized gas MRI, which is commonly used for lung imaging, additional challenges arise due to the nature of the technique. Since repeated scans are often impractical, accounting for and mitigating the impact of noise during the initial acquisition becomes essential for ensuring diagnostic quality and accurate analysis [11].

A widely recognized guideline for determining whether an object can be reliably detected in a noisy image is the Rose Signal-to-Noise Ratio (SNR) criterion [12]. Originally developed in the context of electronic imaging systems, the Rose criterion states that an SNR of approximately five or greater is typically required for a human observer to distinguish an object from the background with high confidence. This rule, though empirical, has been broadly adopted in medical imaging to assess the detectability of anatomical structures in low-contrast or noisy environments.

The Rose SNR is defined as the ratio between the signal difference (contrast) and the standard deviation of the background noise:(1)$${SNR}_{Rose}=\frac{\mu_{signal}-\mu_{background}}{\sigma_{noise}}$$
where $\mu_{signal}$ is the mean signal intensity, $\mu_{background}$ is the mean background intensity, and $\sigma_{noise}$ is the standard deviation of the noise in the background region.

In the context of lung MRI, particularly with hyperpolarized gases, achieving a sufficient Rose SNR is critical because of the inherently low signal and high noise susceptibility due to rapid acquisition and patient-related variability. Adhering to the Rose SNR rule ensures that segmented lung structures are not only mathematically accurate but also visually discernible and clinically interpretable. Therefore, robust segmentation models must perform reliably even in low-SNR regimes to support confident diagnosis and quantitative analysis.

Convolutional Neural Networks (CNNs) have long been the cornerstone for medical image segmentation tasks due to their ability to effectively capture spatial hierarchies in images [13]. Their widespread adoption is largely attributed to their performance in segmenting complex structures with relatively high accuracy. Models such as U-Net [14] and fully convolutional networks (FCNs) [15] have shown remarkable success in segmenting organs and tissues across various modalities, including MRI, CT, and ultrasound images. By leveraging convolutional layers, these models are able to learn spatial features and local dependencies, making them particularly suited for tasks where the structural composition of an image is important.

In the case of lung imaging and hyperpolarized gas MRI, CNNs have demonstrated strong capabilities in delineating airways, lung parenchyma, and other anatomical structures [16,17,18,19,20,21]. The hierarchical feature extraction process allows CNNs to capture both fine-grained details and broader contextual information, which is crucial for accurate segmentation.

Despite their effectiveness, CNN-based models exhibit several limitations when the input images are not in optimal conditions. The local nature of convolution operations makes CNNs highly sensitive to noise and artifacts, which can significantly degrade segmentation accuracy. For instance, when dealing with noisy MRI images, such as those acquired through hyperpolarized gas MRI, CNNs may struggle to distinguish between true anatomical structures and noise-induced artifacts.

Since CNNs rely on small, local receptive fields, they tend to focus on specific regions of the image rather than capturing global context [22]. In the presence of noise, this can lead to the misidentification of boundaries or the omission of important structures. Moreover, CNNs may require extensive preprocessing or denoising techniques to ensure that segmentation results remain reliable, increasing the overall complexity of the workflow. These shortcomings highlight the need for alternative models that are better equipped to handle noise, artifacts, and complex anatomical variations.

Vision Transformers (ViTs) [23], including models like SegFormer [24,25,26,27], have emerged as a powerful alternative to CNNs, particularly in scenarios where image quality is compromised. Unlike CNNs, Vision Transformers leverage attention mechanisms to capture global context in an image, making them more robust to noise and less reliant on local pixel values. This means that while CNNs primarily focus on nearby pixels, ViTs can directly relate distant regions of an image, allowing them to capture long-range dependencies and contextual information.

This difference is particularly advantageous in noisy environments when local pixel values are corrupted and CNNs struggle because their perception is limited to local neighborhoods. In contrast, ViTs can “see” beyond the noisy region and integrate cleaner contextual cues from other parts of the image. As a result, they maintain a holistic understanding of spatial structure, making their segmentations more robust to noise and less dependent on precise local detail [28].

Several pioneering ViT-based segmentation architectures have been developed to address the limitations of traditional CNN approaches. SETR [29] (Segmentation Transformer) was among the first pure-ViT segmentation models, treating segmentation as a sequence prediction problem and demonstrating the feasibility of transformer-based approaches for dense prediction tasks. Segmenter [30] further refined this concept by providing a unified framework for semantic segmentation. Mask2Former [31] extended ViT capabilities to unified instance, panoptic, and semantic segmentation across multiple datasets, including COCO and Cityscapes.

In the medical imaging domain, specialized ViT architectures have shown particular promise. TransUNet [32] pioneered the combination of ViT encoders with CNN decoders in a U-Net-style architecture, specifically targeting medical image segmentation tasks in CT and MRI. Swin-Unet [33] combined the Swin Transformer as an encoder with U-Net-like decoder architectures, demonstrating strong performance on biomedical images. UNETR [34] (UNEt TRansformers) further advanced this field by integrating transformers into the classic U-Net framework for volumetric medical image segmentation.

Among these developments, SegFormer [24] emerged as one of the most promising architectures due to its efficiency and robustness characteristics. Developed by NVIDIA Research, SegFormer has gained recognition as one of the well-established ViT models that is particularly suited for handling challenging imaging conditions, including noise-degraded scenarios.

SegFormer, a lightweight transformer model designed specifically for segmentation tasks, offers several advantages over CNNs. It removes the need for positional encodings, allowing for more flexible feature extraction across scales. This multi-scale feature aggregation enables SegFormer to capture both fine details and global structure, which is crucial for accurate segmentation in noisy images, where local context alone may not be sufficient.

SegFormer’s attention mechanism allows it to focus on the most important regions of the image while disregarding noise and irrelevant artifacts [35]. This capability gives it a distinct edge over CNN-based models in handling noisy MRI data, such as hyperpolarized gas MRI, where imperfections in image quality are common due to motion artifacts, thermal noise, and other factors. As Vision Transformer models are less explored in the context of hyperpolarized gas MRI segmentation, our work aims to fill this gap by comparing the performance of SegFormer with traditional CNN-based methods.

Parallel to these developments, foundation models—large, pre-trained, often promptable architectures—have gained significant attention in medical image segmentation. The Segment Anything Model (SAM), originally trained on general-domain images, exhibits impressive zero-shot abilities but shows significant performance decline when applied directly to medical images with weak boundaries or low contrast. This underperformance has motivated specialized adaptations such as MedSAM, which undergoes fine-tuning on over 1.5 million medical image–mask pairs across modalities. MedSAM demonstrates strong internal and external validation, outperforming both vanilla SAM and specialist models in several tasks. Subsequent variants like MCP-MedSAM and EMedSAM have further refined efficiency and real-time usability, with EMedSAM showing particularly strong results on organ and lung segmentation benchmarks through its compact encoder and tailored processing. This paper aims to conduct a comparative study between the performance of SegFormer, a Vision Transformer-based model, and CNN-based models in the segmentation of hyperpolarized gas MRI scans in the presence of noise. By evaluating the segmentation accuracy under varying levels of noise and artifacts, we seek to identify the strengths and weaknesses of both approaches. While CNNs have been effective for segmentation tasks in the past, their sensitivity to noise may hinder their performance in real-world clinical settings, where image quality can be suboptimal.

Despite the promising advances in foundation models, we did not include them in our comparative analysis due to their substantially different computational complexity, which would not provide a fair comparison framework. Foundation models like MedSAM operate with vastly different parameter scales and computational requirements compared to traditional CNNs and ViTs, making direct performance comparisons inherently biased toward models with greater computational resources. To ensure a meaningful and equitable comparison, we focused our investigation on architectures with comparable computational footprints—specifically established CNN architectures and emerging ViT-based models—allowing us to isolate architectural advantages rather than benefits stemming from computational scale differences. We hypothesize that Vision Transformer-based models—particularly SegFormer—will exhibit greater robustness and maintain higher segmentation accuracy than CNN-based models as image noise increases due to their ability to capture global context and reduce reliance on localized features.

## 2. Materials and Methods

We gathered MRI data from 205 participants, consisting of 22 healthy individuals, 26 with COPD, 90 with asthma, and 67 with long-COVID-19. To improve the reliability of image quantification, particularly for metrics like VDP, only eight central slices were used, excluding the most superior and anterior slices, which are often irregularly shaped and represent low lung volumes. In total, 1640 2D slices were extracted, each with a resolution of 128 × 128 pixels. The segmentation of these images was initially performed using MATLAB R2021b, utilizing a hierarchical K-means clustering algorithm. Proton and hyperpolarized gas MRI slices were aligned using a landmark-based affine registration method to ensure accurate correspondence between the two modalities.

The dataset was balanced across all participant groups to avoid class imbalance and was divided into training (80%, n = 1312), validation (10%, n = 164), and testing (10%, n = 164) sets. To prevent data leakage, each participant’s data was included in only one of these splits.

The original hyperpolarized gas MRI data were acquired as 2D coronal slices, so using 2D slices in our analysis was a natural choice. This approach ensured consistency with the acquisition protocol and allowed for direct evaluation of model performance on clinically relevant imaging data. Analyzing 2D slices also facilitated comparison across participants and disease groups, as each slice corresponded to a standardized anatomical plane.

For the deep learning-based segmentation, we evaluated a range of convolutional and transformer-based architectures. U-Net, a commonly used architecture in biomedical imaging, was selected for its encoder–decoder structure that captures contextual features during downsampling and recovers spatial details through upsampling. U-Net’s distinctive skip connections between encoder and decoder layers help preserve spatial resolution, making it highly effective for precise segmentation.

We also explored the Feature Pyramid Network (FPN) architecture, which builds a multi-scale feature representation by combining low-resolution, semantically rich features with high-resolution, spatially precise features through a top-down pathway with lateral connections. This design enhances the model’s ability to detect and segment structures of varying sizes.

As backbones for U-Net and FPN, we implemented both VGG and ResNet models. VGG16 and VGG19 are deep convolutional networks composed of sequential 3 × 3 convolution layers and max-pooling, offering a simple and uniform architecture for feature extraction. In contrast, ResNet152 introduces residual connections that allow gradients to bypass certain layers, facilitating the training of deeper networks and improving performance on complex segmentation tasks.

In addition to CNN-based models, we employed SegFormer, a Vision Transformer-based model designed for efficient and scalable semantic segmentation. SegFormer uses Mix Transformer (MiT) backbones, which extract multi-scale features via hierarchical attention mechanisms without relying on positional encodings. We evaluated three variants, MiT-B0, MiT-B3, and MiT-B5, which represent increasing model capacities. MiT-B0 is the lightest and fastest variant, MiT-B3 offers a balance between accuracy and efficiency, and MiT-B5 provides the highest segmentation performance at a greater computational cost. These configurations allowed us to assess trade-offs between computational demand and accuracy across diverse conditions.

All models were trained using the Adam optimizer and a Dice Cross Entropy Loss function, which is well-suited for addressing class imbalance in segmentation tasks. Early stopping was used to terminate training when the validation loss plateaued, preventing overfitting. To assess model robustness, Gaussian noise [36] was added to the test set at various levels, defined by standard deviations, to simulate the reduced image quality often encountered in clinical practice. This noise was introduced only during testing to evaluate the models’ generalization ability and their performance in noisy, real-world scenarios without compromising the integrity of training. To simulate real-world imaging conditions, we introduced Gaussian noise at four pre-defined levels: no noise (std = 0.0), low noise (std = 0.05), medium noise (std = 0.15), and high noise (std = 0.25). This enabled us to systematically examine how each model performs as image quality degrades due to noise.

To quantify the level of image degradation introduced by noise, we calculated the average signal-to-noise ratio (SNR) for the central slice across all participants, assuming that the SNR patterns observed in central slices are representative of the entire slice stack. For hyperpolarized gas MRI, under noise-free conditions, the average SNR was 74, indicating excellent image quality. When Gaussian noise with a standard deviation of 0.05 was applied, the SNR dropped to approximately 19. Medium noise (std = 0.15) further reduced the SNR to 9.6, while high noise (std = 0.25) lowered it to 4.5. According to the Rose criterion, an SNR of 5 or greater is generally required for reliable object detectability in noisy images. Notably, the high noise level in our experiments fell below this threshold, providing a meaningful scenario to test the robustness of segmentation models under clinically challenging conditions.

For proton MRI, the signal-to-noise ratio followed a similar degradation pattern under the same noise conditions. Under noise-free conditions, the average SNR for proton MRI was 59, representing good image quality, though lower than that observed in hyperpolarized gas MRI. When low-level Gaussian noise (std = 0.05) was introduced, the SNR decreased to 16.23. Medium noise (std = 0.15) further reduced the SNR to 6.84, while high noise (std = 0.25) resulted in an SNR of 3.86. Notably, the high noise condition for proton MRI fell below the Rose criterion threshold of 5, indicating that reliable object detection becomes significantly more challenging under these conditions. This systematic degradation in both imaging modalities provided a comprehensive framework for evaluating model robustness across varying image quality scenarios commonly encountered in clinical practice.

The effects of progressive noise addition on image quality are visually demonstrated in Figure 1 for hyperpolarized gas MRI and Figure 2 for proton MRI.

## 3. Results

To evaluate the segmentation performance of deep learning models on hyperpolarized gas MRI scans under varying noise conditions, we focused on two well-established architectures: Feature Pyramid Networks (FPNs) and U-Net. Both architectures are widely used in medical image segmentation for their effectiveness in capturing multiscale contextual information. For each, we tested a range of backbones, including CNN-based models (VGG16, VGG19, ResNet-152) and Vision Transformer (ViT)-based models (MiT-B0, MiT-B3, MiT-B5). This setup allowed us to directly compare the capabilities of conventional convolutional approaches with transformer-based alternatives.

We assessed model performance using four standard evaluation metrics for segmentation: Dice Similarity Coefficient (DSC), Average Hausdorff Distance (Avg HD), Hausdorff Distance at the 95th percentile (HD95), and XOR Error. As the Shapiro–Wilk tests indicated violations of normality (*p* < 0.05), we employed the non-parametric Friedman test to assess statistical differences across models. When significant differences were detected, post-hoc pairwise comparisons with Bonferroni correction were performed to identify the specific group differences.

In the following sections, we present a detailed analysis of each architecture’s performance, beginning with Feature Pyramid Networks and followed by U-Net. The performance distributions and statistical significance testing (*p*-values) for all model comparisons across different noise conditions and evaluation metrics are comprehensively documented in the Appendix A.

### 3.1. Proton MRI


**No Noise Condition**


Under the no-noise condition (std = 0.0), all models demonstrated strong segmentation performance on proton MRI scans. Across both FPN and U-Net architectures, Vision Transformer-based models (MiT backbones) consistently outperformed or matched their CNN-based counterparts in all metrics.

For the FPN architecture, the FPN_mit_b5 model achieved the highest Dice Similarity Coefficient (DSC) and the lowest error values across HD95, Avg HD, and XOR metrics, showing significantly better performance than FPN models using VGG16, VGG19, and ResNet152 backbones (*p* < 0.001 in multiple pairwise comparisons). Notably, FPN_mit_b0 and FPN_mit_b3 also showed competitive performance, reinforcing the robustness of the MiT-based backbones in clean imaging conditions.

Similarly, among U-Net models, the transformer-based versions outperformed CNN-based backbones. UNet_mit_b5 showed top performance in DSC and produced the lowest XOR and distance-based errors among all variants. The performance gap between ViT-based and CNN-based models was particularly visible in metrics sensitive to boundary accuracy, such as HD95 and Avg HD. The performance distributions of FPN and U-Net models under the no-noise condition are shown in Appendix A.


**Low Noise Condition**


When subjected to low Gaussian noise (std = 0.05), all models exhibited some degree of performance degradation compared to the no-noise condition. However, ViT-based backbones within both FPN and U-Net architectures demonstrated greater resilience to noise, maintaining higher accuracy and lower error metrics than their CNN-based counterparts.

Within the FPN architecture, FPN_mit_b5 remained the top-performing model, achieving the highest Dice coefficient and the lowest HD95, Avg HD, and XOR scores. The differences between ViT-based and CNN-based FPN models were statistically significant across most metrics (*p* < 0.001). Notably, FPN_resnet152 showed more robust behavior than VGG-based FPNs but still underperformed relative to MiT backbones.

In the U-Net family, UNet_mit_b5 again led in performance, followed closely by UNet_mit_b3, both showing stronger resistance to noise than ResNet- and VGG-based backbones. The impact of noise was more pronounced in CNN-based U-Nets, particularly with UNet_vgg16 and UNet_vgg19, which showed increased variability and degraded performance in HD95 and XOR metrics. Transformer-based U-Nets maintained tighter distributions and lower outlier presence, indicating more stable segmentation outputs under slight image degradation. Appendix A illustrate the segmentation performance of FPN and U-Net models under low noise (std = 0.05), highlighting the comparative resilience of transformer-based models.


**Medium Noise Condition**


At the medium noise level (std = 0.15), segmentation performance declined more noticeably across all models, yet the gap between transformer-based and CNN-based backbones widened further. The presence of moderate Gaussian noise exposed the limitations of convolutional models, particularly in boundary-sensitive metrics.

Within the FPN architecture, ViT-based models (notably FPN_mit_b5 and FPN_mit_b3) continued to demonstrate superior performance. They maintained relatively high Dice scores and significantly lower HD95, Avg HD, and XOR values compared to their CNN counterparts. In contrast, FPN_vgg19 and FPN_vgg16 suffered considerable degradation, with wider performance variability and increased outliers, especially in distance-based metrics.

Transformer-based models also showed more compact metric distributions, indicating more consistent segmentations despite the presence of noise. FPN_resnet152 showed better resilience than VGG-based backbones but still lagged behind the MiT-based variants.

For the U-Net architecture, the pattern remained consistent. UNet_mit_b5 and UNet_mit_b3 offered the best robustness, achieving higher Dice scores and lower HD95 and XOR errors. The CNN-based U-Nets, particularly those with VGG backbones, exhibited degraded performance, with higher spread and more extreme values in HD95 and XOR metrics. Interestingly, UNet_resnet152 performed better than the VGG-based models but was still significantly outperformed by the MiT-based ones in almost all metrics (*p* < 0.001). The effects of medium-level noise (std = 0.15) on FPN and U-Net models are depicted in Appendix A, demonstrating the growing divergence in performance between CNN- and transformer-based models.


**High Noise Condition**


In the high noise condition (std = 0.25), segmentation performance deteriorated significantly across all models, highlighting the challenges posed by substantial image degradation. The impact of noise was especially pronounced in CNN-based models, which showed sharp drops in Dice scores and substantial increases in HD95, Avg HD, and XOR metrics.

For the FPN architecture, the transformer-based models, particularly FPN_mit_b5, continued to show remarkable robustness. Despite the high noise level, they maintained relatively high segmentation accuracy with less variance in all four evaluation metrics. In contrast, FPN_resnet152, FPN_vgg19, and FPN_vgg16 suffered extreme performance drops. Their Dice coefficients plummeted, and both HD95 and XOR errors increased dramatically, with visibly wider distributions and more frequent outliers, suggesting unstable segmentation outputs.

Similar trends were observed in the U-Net architecture. UNet_mit_b5 and UNet_mit_b3 again showed superior performance, with significantly lower error metrics compared to their CNN-based counterparts. UNet_resnet152, UNet_vgg19, and UNet_vgg16 demonstrated severe degradation, especially in boundary-sensitive metrics like HD95, where large outlier values indicated the models’ failure to localize structures accurately under noise.

The ViT-based models, while affected by noise, consistently outperformed CNN-based backbones in both FPN and U-Net setups. Their ability to retain meaningful spatial representations even in noisy environments underscores the advantage of global attention mechanisms over local convolutional filters in handling high-variance inputs.

Appendix A present the segmentation outcomes under high noise (std = 0.25), where the superiority of ViT-based backbones over CNN-based ones becomes increasingly evident.

### 3.2. Hyperpolarized Gas MRI


**No Noise Condition**


In the absence of noise, all models performed strongly on the hyperpolarized gas MRI segmentation task, with minimal errors across all metrics. As with proton MRI, the ViT-based models exhibited slightly better performance, especially in metrics sensitive to shape and boundary accuracy.

For the FPN architecture, FPN_mit_b5 achieved the highest Dice score and lowest values in HD95, Avg HD, and XOR, outperforming both VGG- and ResNet-based backbones. FPN_mit_b0 and FPN_mit_b3 also maintained solid performance with consistent results across samples. The CNN-based backbones (particularly FPN_vgg19) showed a small but noticeable drop in accuracy, particularly in XOR and HD95 metrics, where longer tails and increased variability indicated less consistent boundary adherence.

In the U-Net architecture, UNet_mit_b5 was again the top performer across nearly all metrics, closely followed by UNet_mit_b3. These transformer-based U-Nets showed tighter distributions in HD95 and XOR, suggesting more stable and accurate delineation of lung regions. CNN-based models such as UNet_vgg16 and UNet_resnet152 showed acceptable performance but with slightly wider error distributions, particularly in Avg HD. Appendix A visualize model performance for FPN and U-Net architectures, respectively, in clean gas MRI conditions (std = 0.0).


**Low Noise Condition**


Under the low noise condition (std = 0.05), all models experienced a slight performance decline, but the transformer-based models again demonstrated superior resilience and consistency compared to their CNN-based counterparts.

In the FPN architecture, FPN_mit_b5 and FPN_mit_b3 continued to lead across all metrics. These models maintained high Dice scores and low errors in HD95, Avg HD, and XOR, with narrow distributions indicating reliable performance. In contrast, CNN-based models such as FPN_vgg19 and FPN_vgg16 showed increased variability in boundary-sensitive metrics, particularly HD95 and XOR, reflecting their higher sensitivity to noise artifacts.

FPN_resnet152 offered slightly better stability than the VGG-based FPNs but still lagged behind the MiT-based models. Notably, pairwise comparisons (*p* < 0.001 in most metrics) confirmed the statistically significant performance advantage of transformer-based FPNs under noisy conditions.

Within the U-Net architecture, UNet_mit_b5 once again achieved the best overall performance, followed by UNet_mit_b3. These models displayed strong Dice accuracy and minimal degradation in spatial metrics. The CNN-based U-Nets, particularly those with VGG backbones, were more adversely affected by noise, showing wider spreads in Avg HD and XOR. Under low noise conditions (std = 0.05), the segmentation results for FPN and U-Net models are displayed in Appendix A.


**Medium Noise Condition**


With the introduction of medium-level noise (std = 0.15), model performance began to diverge more significantly, revealing clear differences in robustness across architectures and backbones. While all models showed some performance degradation, transformer-based backbones remained notably more stable and accurate than their CNN counterparts.

In the FPN architecture, FPN_mit_b5 retained the best overall performance across all metrics. Dice scores dropped slightly but remained high, and the HD95, Avg HD, and XOR values remained significantly lower than those of CNN-based FPN models (*p* < 0.001 in most pairwise tests). The CNN-based models, particularly FPN_vgg19, showed marked declines in segmentation quality, with wider spread and more outliers in HD95 and XOR, indicating increasing sensitivity to noise.

FPN_resnet152 maintained moderate performance but still failed to match the consistency and accuracy of MiT-based variants. The increased metric dispersion in CNN models highlights their limited ability to generalize when image quality is moderately degraded.

In the U-Net architecture, UNet_mit_b5 and UNet_mit_b3 again demonstrated greater resilience. They showed minimal impact in Dice and spatial accuracy metrics compared to the baseline noise-free condition. In contrast, UNet_vgg16, UNet_vgg19, and UNet_resnet152 experienced a notable increase in error variance, with frequent outliers in HD95 and XOR indicating unstable segmentation results. The segmentation accuracy of FPN and U-Net models at a medium noise level (std = 0.15) is shown in Appendix A, revealing model-specific differences in robustness.


**High Noise Condition**


At the highest noise level (std = 0.25), all models experienced a significant decline in segmentation performance, with CNN-based architectures being especially vulnerable. The results revealed large variability, increased error rates, and frequent outliers, particularly in boundary-sensitive metrics such as HD95 and XOR.

For the FPN architecture, MiT-based models—especially FPN_mit_b5 and FPN_mit_b3—showed clear robustness and outperformed all CNN-based backbones. While performance degradation was evident, these models preserved higher Dice scores and maintained more compact distributions across HD95 and Avg HD. In contrast, FPN_vgg16 and FPN_vgg19 suffered substantial accuracy drops, with highly dispersed error metrics and elevated XOR values, indicating poor structural consistency in segmentation masks.

FPN_resnet152 performed better than the VGG variants but was still significantly less stable than transformer-based FPNs. Notably, statistical tests confirmed consistent significant differences (*p* < 0.001) in all four metrics between ViT-based and CNN-based FPNs under high noise.

The U-Net architecture exhibited a similar trend. UNet_mit_b5 and UNet_mit_b3 maintained a noticeable edge over CNN counterparts, producing higher-quality segmentations despite increased noise. Their Dice coefficients remained elevated, and their HD95 and XOR values, while increased, remained within a tighter range. Conversely, UNet_vgg19 and UNet_vgg16 showed wide performance dispersion, with several cases of segmentation failure reflected by outlier values and high mean errors across metrics.

Ultimately, this condition highlights the critical advantage of Vision Transformer-based models in handling severe image degradation. Their attention-based mechanisms enable more effective modeling of global context and resilience to local distortions, making them more suitable for real-world deployment in noisy clinical imaging scenarios like hyperpolarized gas MRI. Appendix A summarize the segmentation performance under high noise (std = 0.25), where ViT-based architectures demonstrate enhanced stability over CNN-based alternatives.

### 3.3. Effect Size Analysis

Effect sizes were calculated using Cohen’s d to quantify the practical significance of performance differences between ViT and CNN models. Cohen’s d represents the standardized mean difference between groups, with values interpreted as negligible (|d| < 0.2), small (0.2 ≤ |d| < 0.5), medium (0.5 ≤ |d| < 0.8), or large (|d| ≥ 0.8) according to conventional thresholds [37]. Ninety-five percent confidence intervals were calculated using the standard error of Cohen’s d to assess the precision of effect size estimates. For metrics where lower values indicate better performance (Avg HD, HD95, XOR), effect sizes were calculated as CNN minus ViT values, such that positive effect sizes consistently indicate ViT superiority across all metrics.

Effect size analysis revealed a systematic pattern of increasing ViT superiority as noise levels increased (Table 1, Appendix A). Under noise-free conditions, effect sizes were negligible for both imaging modalities, with CNNs showing slight advantages in proton MRI (d = −0.11 to −0.14 across metrics) and equivalent performance in hyperpolarized gas MRI (d = 0.04 to 0.05).

As noise increased, ViT models demonstrated progressively larger advantages. At high noise levels, ViT models achieved large effect sizes in proton MRI (d = 0.7192 to 0.93 across metrics) and medium effect sizes in hyperpolarized gas MRI (d = 0.52 to 0.57). The most substantial differences were observed for the XOR metric in proton MRI under high noise conditions (d = 0.93, 95% CI: 0.88–0.9879), indicating a large practical advantage for ViT-based segmentation.

The dose–response relationship between noise level and ViT superiority was consistent across both imaging modalities, though the magnitude of effect was greater in proton MRI. This pattern supports the hypothesis that ViT models’ global attention mechanisms provide superior robustness to image degradation compared to the local receptive fields of CNN architectures.

## 4. Discussion

This study investigated the robustness of deep learning segmentation models under varying levels of Gaussian noise in both proton and hyperpolarized gas MRI, with a specific focus on comparing traditional CNN-based architectures and modern Vision Transformer (ViT)-based approaches. The results across all metrics consistently demonstrated that ViT-based models, particularly those using the SegFormer backbone (MiT-B5 and MiT-B3), outperformed CNN counterparts under both clean and noisy conditions. This was evident in both FPN and U-Net architectural families, reinforcing the hypothesis that global context modeling through attention mechanisms offers significant advantages in the presence of image degradation.

The findings are particularly impactful in the context of hyperpolarized gas MRI, where noise and motion artifacts are an intrinsic challenge due to breath-hold requirements, lower signal-to-noise ratios (SNRs), and the limited time window available for image acquisition. Unlike conventional proton MRI, hyperpolarized gas MRI captures fleeting moments of gas distribution during a single breath-hold, making it especially sensitive to noise and variations in image quality. Therefore, segmentation algorithms deployed in this context must be highly robust to ensure consistent and accurate delineation of lung regions across varying acquisition conditions.

This performance trend is visually evident in the qualitative segmentation results shown in Figure 3, Figure 4, Figure 5 and Figure 6. Specifically, Figure 3 and Figure 5 illustrate the robustness of the ViT-based UNet with an MiT-B5 backbone on proton and hyperpolarized gas MRI, respectively, where accurate segmentation is preserved even at high noise levels. In contrast, Figure 4 and Figure 6 highlight the degradation in segmentation quality for the CNN-based UNet with a ResNet-152 backbone, where boundary accuracy and Dice scores drop considerably under increasing noise. These visual examples further support the quantitative findings and reinforce the advantage of transformer-based architectures in noisy clinical environments.

Our results showed that CNN-based models degrade rapidly as noise levels increase. This is particularly problematic in clinical hyperpolarized MRI workflows, where repeat imaging is impractical and model reliability is critical from the first attempt. As illustrated in Figure 7 the average DSC for CNN-based models dropped sharply at high noise levels, whereas ViT-based models maintained better performance and stability. This highlights the potential of transformer-based models for clinical deployment in hyperpolarized gas MRI, where precision and resilience are critical.

This performance gap is clearly visualized in Figure 7, Figure 8, Figure 9, Figure 10, Figure 11, Figure 12, Figure 13 and Figure 14, where Dice Similarity Coefficient, Average Hausdorff Distance, 95th Percentile Hausdorff Distance, and XOR trends across noise levels show that ViT-based models consistently outperform CNN-based counterparts, particularly under medium and high noise conditions.

These findings align with recent studies that have demonstrated the benefits of attention-based architectures in low-quality imaging scenarios. Previous research has noted that CNNs, while effective on clean data, rely heavily on local spatial features and are prone to boundary misidentification in noisy conditions. In contrast, transformers excel at learning long-range dependencies and can leverage context from the entire image to reinforce boundary consistency, which is crucial when anatomical edges are partially occluded or distorted by noise.

Moreover, models like SegFormer are lightweight yet highly expressive, allowing for deployment even in computationally constrained environments. Their multi-scale feature fusion further enhances their capability to handle complex structures like lung lobes, which may vary significantly in appearance between patients and disease states.

These results advocate for a paradigm shift in model selection for medical image segmentation, especially in challenging modalities like hyperpolarized gas MRI. Vision Transformers offer a promising path toward more generalizable, noise-tolerant segmentation solutions that can be used reliably across patient populations and imaging systems.

Additionally, the consistent superiority of ViT-based models in both FPN and U-Net architectures indicates that their advantages are not limited to specific designs but rather stem from their underlying attention-based mechanism. This generalizability opens up opportunities for integrating ViTs into other parts of the medical imaging pipeline, such as motion correction, registration, or multimodal fusion tasks.

While statistical significance indicates that observed differences are unlikely due to chance, effect sizes quantify whether these differences are practically meaningful. The large effect sizes observed under high noise conditions (d > 0.8 for proton MRI) suggest that the choice between ViT and CNN architectures can have a substantial impact in challenging imaging scenarios.

Particularly noteworthy is the finding that ViT superiority emerges only under degraded image conditions. This suggests that the global attention mechanisms of transformers provide a specific advantage for handling noise, artifacts, and non-ideal imaging conditions, rather than representing a universal improvement over CNNs. This has important implications for clinical deployment decisions: while CNNs may suffice for high-quality imaging scenarios, ViT-based models should be preferred when image quality is expected to be suboptimal, such as in rapid acquisition protocols, patient motion scenarios, or when using lower-field MRI systems.

Although this study was conducted using medical images of the lung, the segmentation models and approaches evaluated here are not specific to pulmonary imaging and can be applied to other types of medical images. Vision Transformer-based architectures such as SegFormer are designed to learn global contextual features and are inherently modality- and organ-agnostic. As a result, the observed improvements in robustness and segmentation accuracy under noise are likely to extend to other anatomical structures and imaging modalities, such as abdominal, cardiac, or brain MRI, as well as CT or ultrasound images. The flexibility and generalizability of these deep learning models support their broader adoption in various clinical imaging scenarios where image quality may be compromised. Future studies should explore the adaptation and performance of these segmentation approaches across different organs and modalities to further validate and expand upon the conclusions presented here.

An important limitation of this study is the exclusive use of 2D slice-wise segmentation rather than volumetric 3D approaches. While our 2D analysis was consistent with the original acquisition protocol for hyperpolarized gas MRI (high-resolution 3D isotropic voxel measurements are rarely feasible due to the 16-second breath-hold limitation) and facilitated direct comparison across participants and disease groups, it inherently lacks the volumetric context that 3D segmentation provides. This limitation is particularly relevant in lung imaging, where anatomical structures exhibit complex 3D relationships and pathological features may span multiple slices. 2D segmentation cannot capture inter-slice continuity or leverage spatial coherence across the entire lung volume, potentially leading to inconsistencies between adjacent slices and suboptimal boundary delineation at slice interfaces. Furthermore, clinically relevant metrics such as total lung volume, regional ventilation distribution, and ventilation defect percentage are typically calculated from complete 3D lung volumes rather than individual 2D slices.

The 2D approach also limits the clinical applicability of our findings, as real-world clinical workflows increasingly rely on volumetric analysis for the comprehensive assessment of lung function and pathology. Modern 3D segmentation models can exploit spatial continuity across slices through techniques such as 3D convolutions, volumetric attention mechanisms, and inter-slice consistency constraints, potentially offering superior performance compared to slice-by-slice analysis. Additionally, 3D models may be more robust to noise by leveraging information from neighboring slices to compensate for degraded image quality in individual slices. Future work should extend this comparative analysis to 3D architectures, including recent developments such as SegFormer3D [38] and UNETR [34], to evaluate whether the observed advantages of Vision Transformers over CNNs persist in volumetric segmentation tasks and to assess the clinical impact of transitioning from 2D to 3D segmentation workflows in hyperpolarized gas MRI.

A limitation of this study is that no formal mathematical framework was developed to evaluate the computational robustness or sensitivity of the segmentation models to different types and magnitudes of noise. However, model performance was quantitatively assessed using widely accepted segmentation metrics: DSC, which measures the overlap between predicted and ground truth masks; Avg HD and HD95, which quantify boundary accuracy and the degree of spatial mismatch between segmented regions; and XOR error, which captures the total number of mismatched pixels. These metrics collectively provide a comprehensive assessment of segmentation performance in terms of both region overlap and boundary precision. Future work may benefit from incorporating shape-based or topological analysis frameworks to further elucidate model robustness and segmentation fidelity under challenging imaging conditions.

Additionally, while we conducted preliminary pilot validation with external datasets from McMaster University and received positive feedback regarding model generalizability, formal quantitative evaluation on external datasets with comprehensive metrics will be included in future studies to further validate the robustness and transferability of our findings.

Although this study demonstrates the technical robustness and accuracy of ViT-based segmentation models, we acknowledge that no pilot validation or preliminary integration into real-world clinical workflows was performed as part of this work. Future work should include prospective studies involving clinician users to evaluate the real-world benefits, user experience, and potential impact on clinical decision-making and workflow efficiency.

## 5. Conclusions and Future Work

In this study, we systematically evaluated the performance and robustness of CNN-based and Vision Transformer (ViT)-based deep learning models for lung segmentation on both proton and hyperpolarized gas MRI scans under varying levels of Gaussian noise. Our findings clearly demonstrate that transformer-based models—particularly those using the SegFormer backbone (MiT variants)—consistently outperform traditional CNN backbones across all tested conditions.

This performance advantage was especially evident in high-noise settings, where CNN models exhibited significant degradation in segmentation accuracy and boundary consistency. In contrast, ViT-based models retained relatively high Dice scores and showed tighter error distributions, confirming their robustness in low-quality imaging environments. These insights are particularly relevant for hyperpolarized gas MRI as well as fluorinated gas lung MRI, where breath-hold requirements, rapid acquisition, and inherently lower SNRs (particularly in the case of ^19^F MRI) make robust segmentation solutions essential.

The results not only validate the potential of Vision Transformers in challenging medical imaging scenarios but also support a broader shift toward attention-based architectures for clinical deployment in segmentation workflows. Their ability to model global context and maintain stability across variable image qualities makes them well-suited for real-world applications, especially in longitudinal studies, multi-center trials, and personalized treatment planning, where repeat scans may be limited.

Effect size analysis revealed that ViT superiority is specifically advantageous under degraded imaging conditions, with large effect sizes (d > 0.8) observed under high-noise scenarios in proton MRI. This quantifies the practical clinical significance of the observed performance differences and provides guidance for architecture selection based on expected image quality.

Building on these findings, future work will explore the integration of foundational segmentation models such as the Segment Anything Model (SAM) and MedSAM. These models, pre-trained on large-scale datasets with general-purpose capabilities, may offer strong zero-shot or few-shot performance and reduce the need for task-specific training data. Evaluating their adaptability to domain-specific applications such as hyperpolarized gas MRI will be crucial in assessing their readiness for clinical use. Foundation models present several compelling advantages over traditional architectures: their extensive pre-training on diverse datasets enables superior generalization across imaging modalities, anatomical structures, and pathological conditions without requiring domain-specific fine-tuning. This is particularly valuable in specialized imaging domains like hyperpolarized gas MRI, where acquiring large, labeled datasets is challenging due to limited patient populations and specialized equipment requirements. Additionally, foundation models support interactive and prompt-based segmentation, allowing clinicians to guide the segmentation process through simple user inputs such as points, bounding boxes, or text descriptions, potentially enabling more flexible and user-friendly clinical workflows.

Furthermore, the promotability of foundation models could address one of the key limitations of traditional CNN and ViT models: their fixed, task-specific nature that requires retraining for different segmentation objectives. In addition, future studies will aim to extend this evaluation to 3D segmentation tasks and explore temporal dynamics in functional MRI studies. Incorporating clinical priors, such as anatomical atlases or physiological constraints, into transformer-based pipelines could further enhance performance and clinical relevance. Hybrid architectures that combine CNN feature extractors with transformer decoders will also be investigated to balance efficiency and accuracy. Furthermore, we plan to explore the use of retrieval-augmented generation (RAG)-based or prompt-tuned transformer models to support interactive or semi-automated clinical workflows. Overall, this work lays a foundation for the next generation of robust, generalizable segmentation models that can meet the demands of noisy, high-variability imaging modalities like hyperpolarized gas MRI.

## Figures and Tables

**Figure 1 bioengineering-12-00808-f001:**
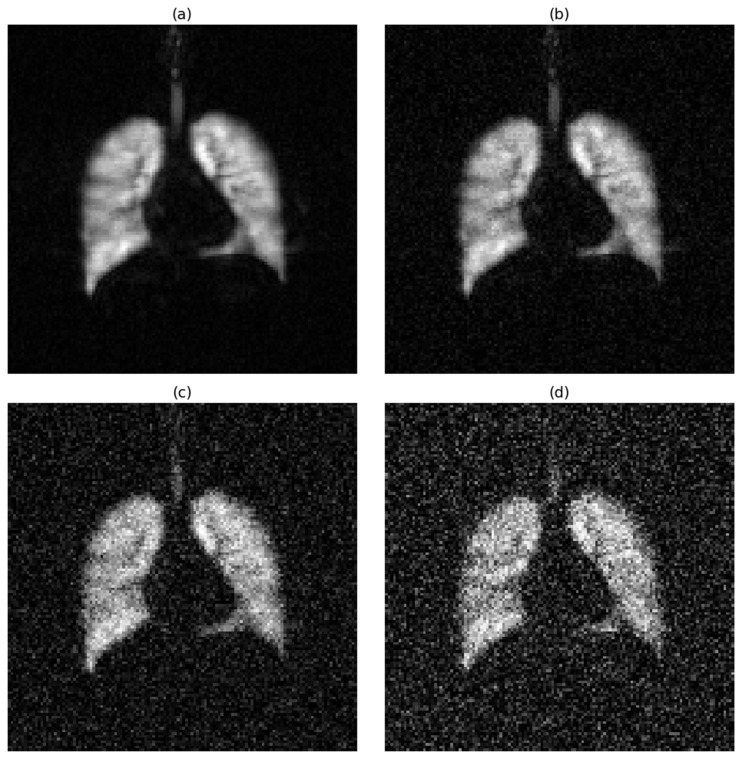
Effect of Gaussian noise on hyperpolarized gas MRI image quality. Representative coronal slice showing progressive image degradation with increasing noise levels: (**a**) noise-free conditions (SNR = 74), (**b**) low noise with a standard deviation of 0.05 (SNR ≈ 19), (**c**) medium noise with a standard deviation of 0.15 (SNR = 9.6), and (**d**) high noise with a standard deviation of 0.25 (SNR = 4.5).

**Figure 2 bioengineering-12-00808-f002:**
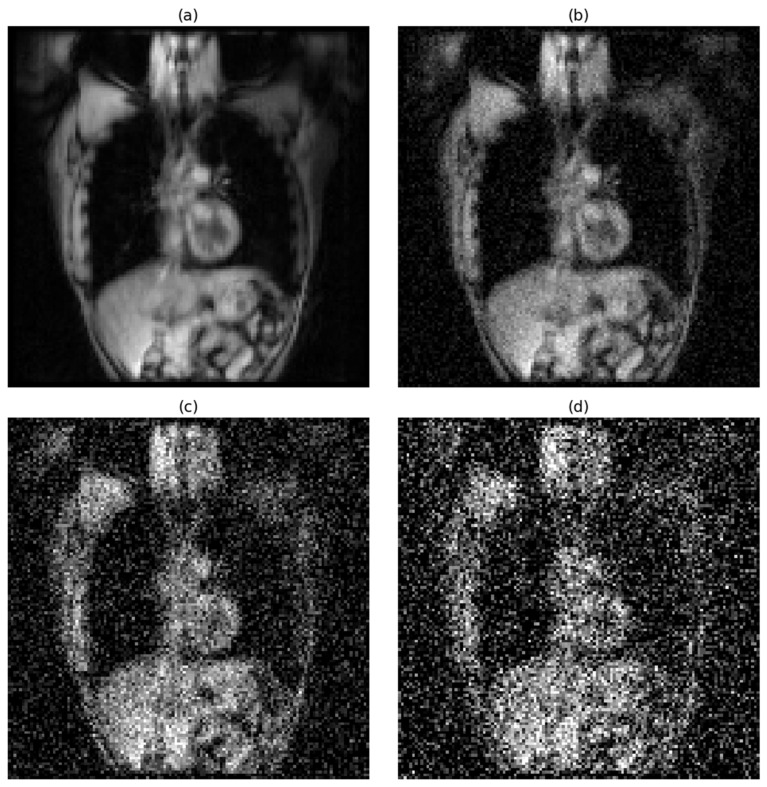
Progressive degradation of proton MRI image quality under varying noise conditions. Representative coronal slice demonstrating the impact of increasing Gaussian noise levels: (**a**) original image without noise (SNR = 59), (**b**) low-level noise with a standard deviation of 0.05 (SNR = 16.23), (**c**) medium-level noise with a standard deviation of 0.15 (SNR = 6.84), and (**d**) high-level noise with a standard deviation of 0.25 (SNR = 3.86).

**Figure 3 bioengineering-12-00808-f003:**
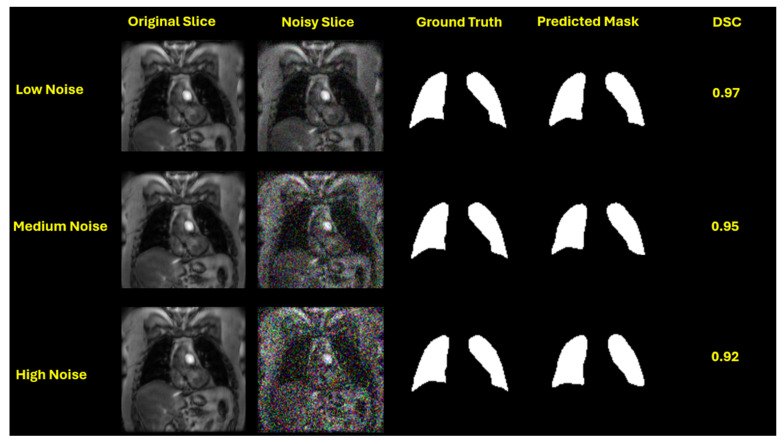
Qualitative segmentation results of UNet with an MiT-B5 backbone on proton MRI under increasing Gaussian noise levels. Each row represents a different noise level—low (std = 0.05), medium (std = 0.15), and high (std = 0.25)—applied during testing. From left to right: original slice, noisy slice, ground truth mask, predicted mask, and Dice Similarity Coefficient (DSC). The model maintains high segmentation accuracy (DSC = 0.97, 0.95, and 0.92, respectively), demonstrating robustness to image degradation.

**Figure 4 bioengineering-12-00808-f004:**
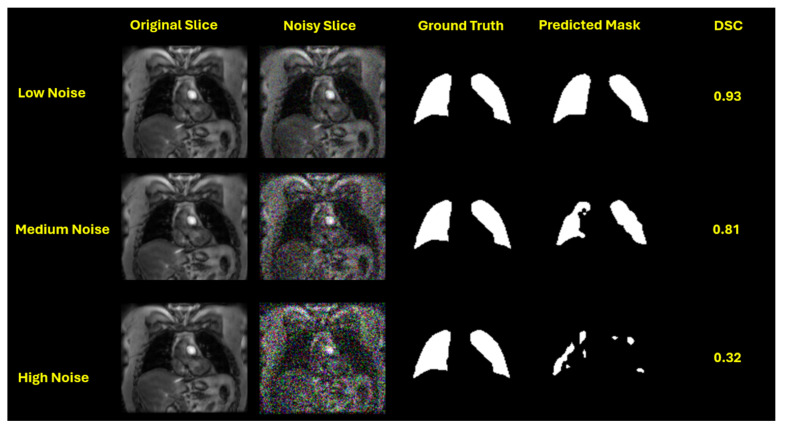
Segmentation performance of UNet with a ResNet-152 backbone on proton MRI slices under increasing Gaussian noise levels. Each row corresponds to a different noise setting: low (std = 0.05), medium (std = 0.15), and high (std = 0.25). From left to right: original slice, noisy slice, ground truth mask, predicted mask, and Dice Similarity Coefficient (DSC). While the model achieves a reasonable DSC of 0.93 under low noise, performance significantly deteriorates at medium (DSC = 0.81) and high noise (DSC = 0.32), revealing its vulnerability to noise-induced artifacts.

**Figure 5 bioengineering-12-00808-f005:**
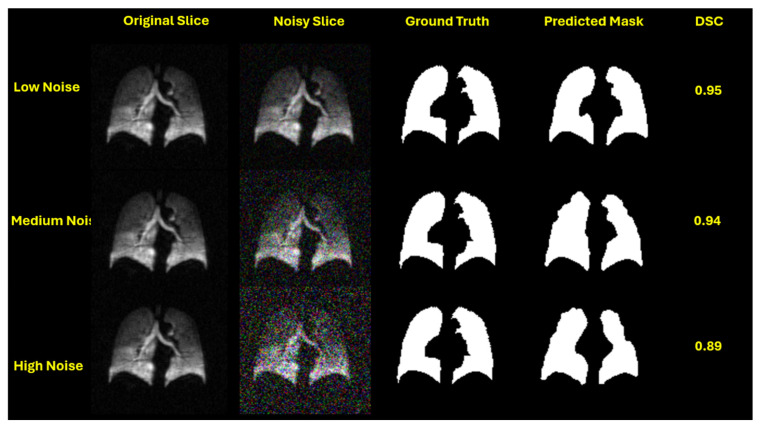
UNet with an MiT-B5 backbone applied to hyperpolarized gas MRI slices under varying Gaussian noise levels. Rows represent low (std = 0.05), medium (std = 0.15), and high noise (std = 0.25) scenarios. Columns show the original slice, noisy input, ground truth mask, predicted segmentation, and corresponding Dice Similarity Coefficient (DSC). The model demonstrates strong robustness, maintaining high segmentation accuracy across all conditions (DSC = 0.95, 0.94, 0.89), highlighting its ability to generalize under noisy clinical imaging settings.

**Figure 6 bioengineering-12-00808-f006:**
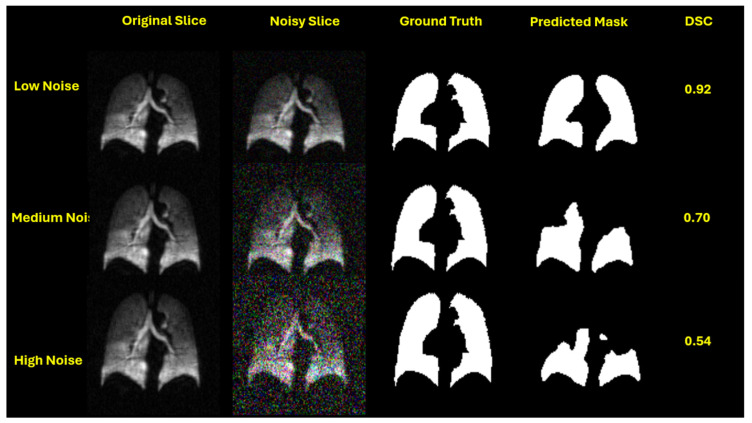
Segmentation results of UNet with a ResNet-152 backbone on hyperpolarized gas MRI slices across increasing Gaussian noise levels. Rows correspond to low (std = 0.05), medium (std = 0.15), and high noise (std = 0.25). From left to right: original image, noisy slice, ground truth mask, predicted mask, and Dice Similarity Coefficient (DSC). While the model performs adequately under low noise (DSC = 0.92), segmentation quality declines substantially at medium (DSC = 0.70) and high noise levels (DSC = 0.54), revealing reduced robustness in challenging conditions.

**Figure 7 bioengineering-12-00808-f007:**
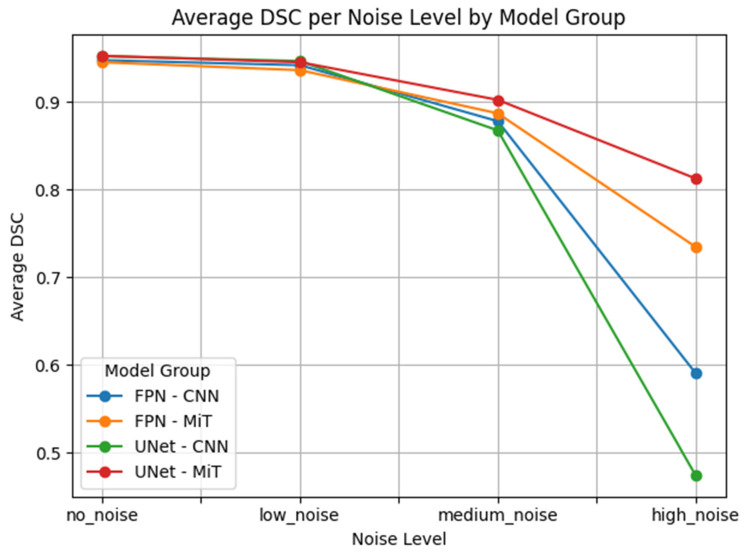
Average Dice Similarity Coefficient (Average DSC) across varying Gaussian noise levels (std = 0.0, 0.05, 0.15, 0.25) for each model group on proton MRI. Vision Transformer (ViT)-based models (FPN–MiT, UNet–MiT) maintain high and stable performance under noise, while CNN-based models (FPN–CNN, UNet–CNN) exhibit significant degradation, particularly at medium and high noise levels, highlighting the superior robustness of transformer-based architectures.

**Figure 8 bioengineering-12-00808-f008:**
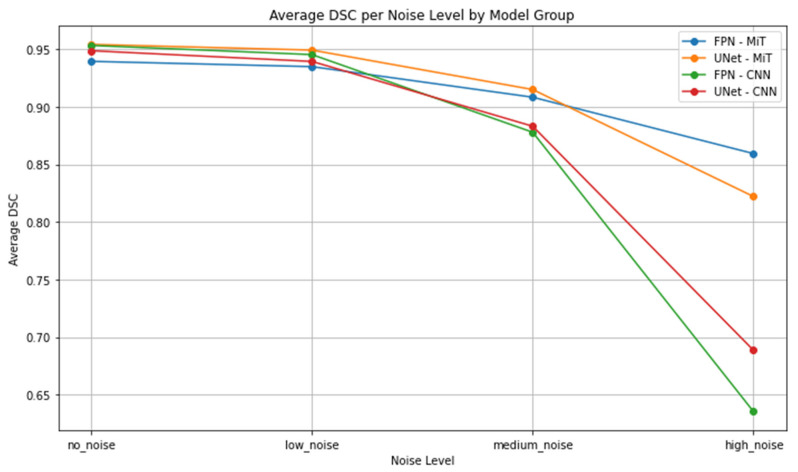
Average Dice Similarity Coefficient (Average DSC) across Gaussian noise levels (std = 0.0, 0.05, 0.15, 0.25) for each model group on hyperpolarized gas MRI. Vision Transformer (ViT)-based models (FPN–MiT, UNet–MiT) demonstrate greater robustness, with a more gradual decline in performance as noise increases. In contrast, CNN-based models (FPN–CNN, UNet–CNN) show a sharper drop in Average DSC at higher noise levels, reflecting their limited resilience in noisy, low-SNR imaging environments.

**Figure 9 bioengineering-12-00808-f009:**
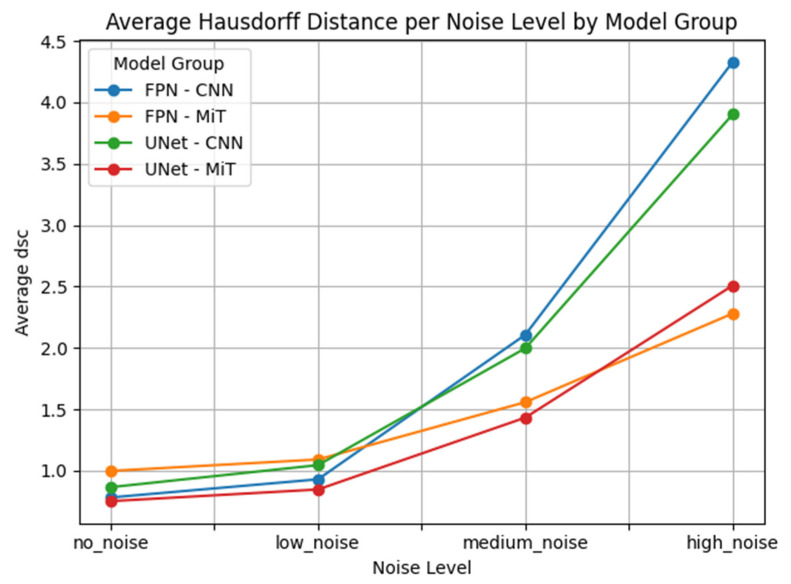
Average Hausdorff Distance (Average HD) across varying Gaussian noise levels (std = 0.0, 0.05, 0.15, 0.25) for each model group on proton MRI. Vision Transformer (ViT)-based models (FPN–MiT, UNet–MiT) maintain high and stable performance under noise, while CNN-based models (FPN–CNN, UNet–CNN) exhibit significant degradation, particularly at medium and high noise levels, highlighting the superior robustness of transformer-based architectures.

**Figure 10 bioengineering-12-00808-f010:**
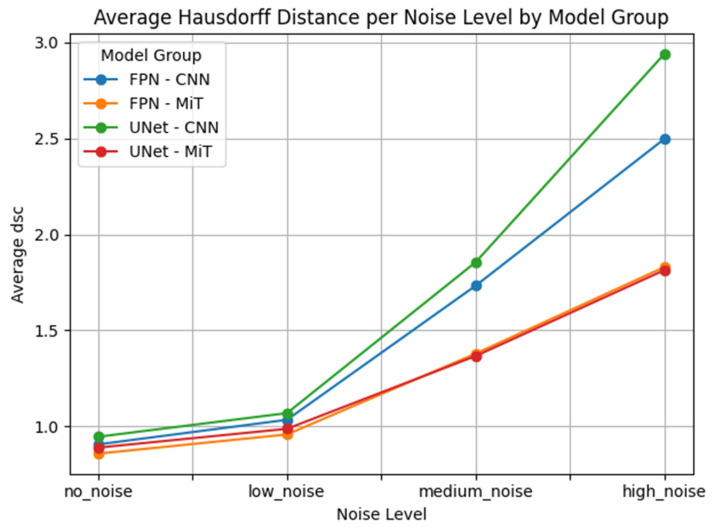
Average Hausdorff Distance (Average HD) across Gaussian noise levels (std = 0.0, 0.05, 0.15, 0.25) for each model group on hyperpolarized gas MRI. Vision Transformer (ViT)-based models (FPN–MiT, UNet–MiT) demonstrate greater robustness, with a more gradual decline in performance as noise increases. In contrast, CNN-based models (FPN–CNN, UNet–CNN) show a sharper drop in DSC at higher noise levels, reflecting their limited resilience in noisy, low-SNR imaging environments.

**Figure 11 bioengineering-12-00808-f011:**
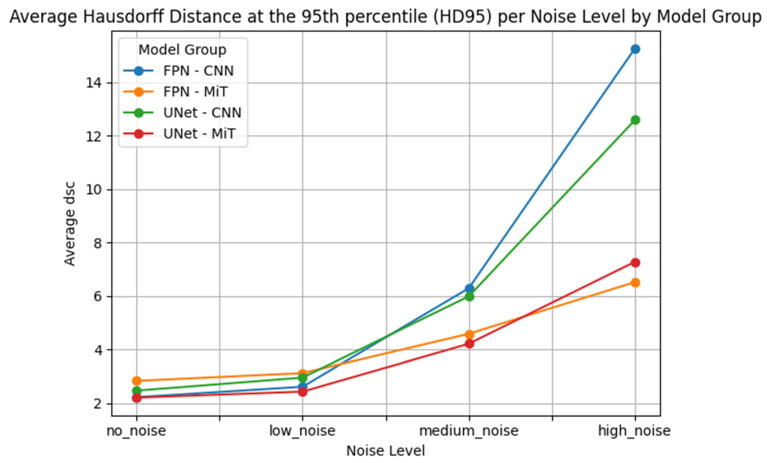
Hausdorff Distance at the 95th percentile (HD95) across varying Gaussian noise levels (std = 0.0, 0.05, 0.15, 0.25) for each model group on proton MRI. Vision Transformer (ViT)-based models (FPN–MiT, UNet–MiT) maintain high and stable performance under noise, while CNN-based models (FPN–CNN, UNet–CNN) exhibit significant degradation, particularly at medium and high noise levels, highlighting the superior robustness of transformer-based architectures.

**Figure 12 bioengineering-12-00808-f012:**
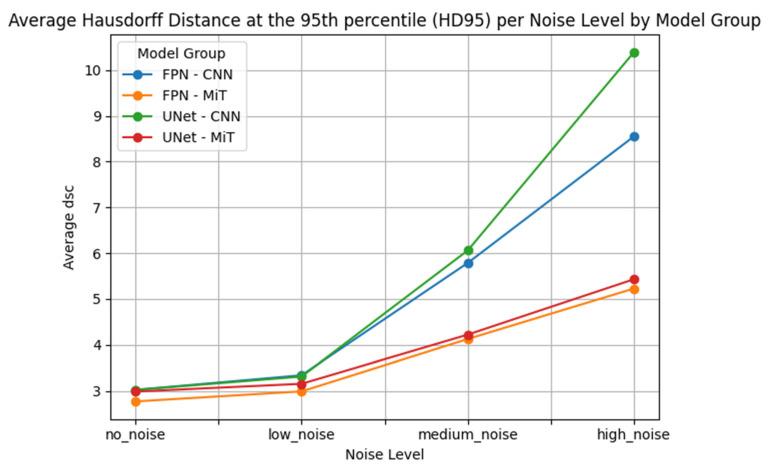
Hausdorff Distance at the 95th percentile (HD95) across Gaussian noise levels (std = 0.0, 0.05, 0.15, 0.25) for each model group on hyperpolarized gas MRI. Vision Transformer (ViT)-based models (FPN–MiT, UNet–MiT) demonstrate greater robustness, with a more gradual decline in performance as noise increases. In contrast, CNN-based models (FPN–CNN, UNet–CNN) show a sharper drop in DSC at higher noise levels, reflecting their limited resilience in noisy, low-SNR imaging environments.

**Figure 13 bioengineering-12-00808-f013:**
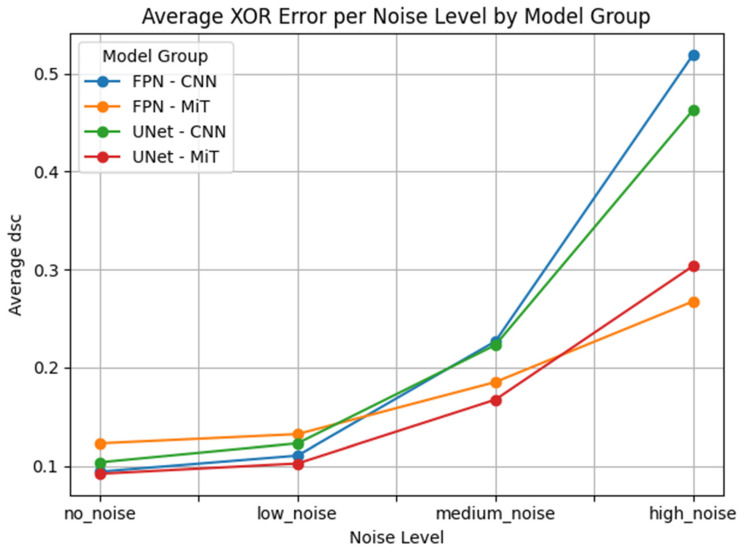
Average XOR error across varying Gaussian noise levels (std = 0.0, 0.05, 0.15, 0.25) for each model group on proton MRI. Vision Transformer (ViT)-based models (FPN–MiT, UNet–MiT) maintain high and stable performance under noise, while CNN-based models (FPN–CNN, UNet–CNN) exhibit significant degradation, particularly at medium and high noise levels, highlighting the superior robustness of transformer-based architectures.

**Figure 14 bioengineering-12-00808-f014:**
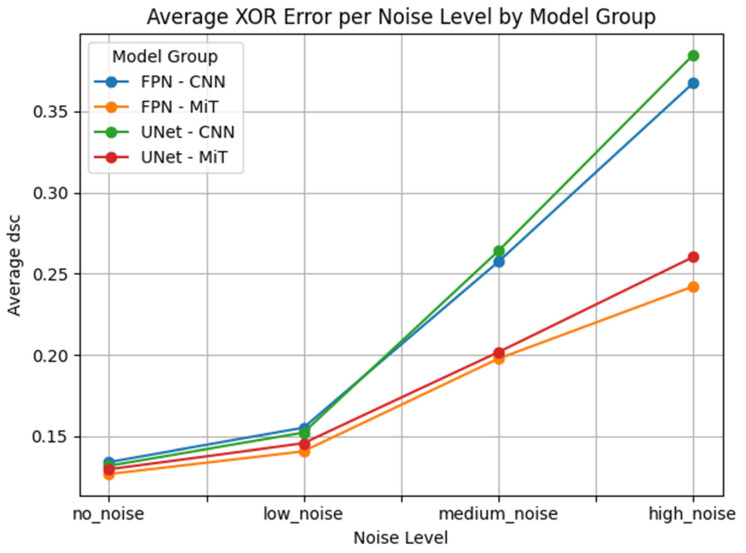
Average XOR error across Gaussian noise levels (std = 0.0, 0.05, 0.15, 0.25) for each model group on hyperpolarized gas MRI. Vision Transformer (ViT)-based models (FPN–MiT, UNet–MiT) demonstrate greater robustness, with a more gradual decline in performance as noise increases. In contrast, CNN-based models (FPN–CNN, UNet–CNN) show a sharper drop in DSC at higher noise levels, reflecting their limited resilience in noisy, low-SNR imaging environments.

**Table 1 bioengineering-12-00808-t001:** Effect sizes (Cohen’s d) for ViT vs. CNN performance across noise conditions. Values represent Cohen’s d with 95% confidence intervals in parentheses. Positive values indicate ViT superiority. * Small effect size (0.2 ≤ |d| < 0.5); ** Medium to large effect size (|d| ≥ 0.5).

*Metric*	Imaging Modality	No Noise	Low Noise	Medium Noise	High Noise
*DSC*	Proton MRI	−0.13 (−0.19, −0.08)	−0.02 (−0.08, 0.02)	0.485 (0.43, 0.53) *	0.85 (0.79, 0.90) **
*DSC*	Hyperpolarized Gas MRI	0.04 (−0.01, 0.10)	0.087 (0.02, 0.14)	0.31 (0.25, 0.36) *	0.52 (0.46, 0.58) **
*Avg HD*	Proton MRI	−0.10 (−0.15, −0.05)	0.01 (−0.04, 0.06)	0.46 (0.41, 0.51) *	0.86 (0.80, 0.91) **
*Avg HD*	Hyperpolarized Gas MRI	0.05 (−0.008, 0.10)	0.08 (0.03, 0.14)	0.40 (0.34, 0.45) *	0.57 (0.51, 0.63) **
*HD95*	Proton MRI	−0.11 (−0.16, −0.06)	−0.01 (−0.06, 0.04)	0.39 (0.34, 0.45) *	0.71 (0.66, 0.77) **
*HD95*	Hyperpolarized Gas MRI	0.05 (−0.00, 0.11)	0.09 (0.03, 0.15)	0.34 (0.28, 0.40) *	0.53 (0.47, 0.59) **
*XOR*	Proton MRI	−0.13 (−0.18, −0.08)	−0.02 (−0.08, 0.02)	0.43 (0.38, 0.49) *	0.93 (0.87, 0.98) **
*XOR*	Hyperpolarized Gas MRI	0.04 (−0.01, 0.10)	0.08 (0.02, 0.14)	0.33 (0.27, 0.39) *	0.56 (0.50, 0.62) **

## Data Availability

The original contributions presented in the study are included in the article/Appendix A, further inquiries can be directed to the corresponding author.

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
