# Peer review of "Robust Segmentation of Lung Proton and Hyperpolarized Gas MRI with Vision Transformers and CNNs: A Comparative Analysis of Performance Under Artificial Noise"

_bioengineering, 2025, doi:10.3390/bioengineering12080808_

Round 1
Reviewer 1 Report
Comments and Suggestions for Authors
This paper considers the segmentation of medical images under different levels of noise. The performance of several different deep learning segmentation models is tested and evaluated under varying Gaussian noise levels. These models include the traditional convolutional neural networks (CNNs) and modern Vision Transformer (ViT)-based models. Results show that ViT-based models consistently outperforms CNN-based models under all noise levels.The paper is generally well written. The results presented are original and may provide insights into the development of methodologies for accurate segmentation of medical images under various noisy conditions. However, the following issues exist in the paper and need to be addressed before it can be accepted for publication.
1. Why only medical images of lungs are considered in the study? Can the conclusions of the paper be extended to other types of medical images? A discussion is needed in the text regarding this issue.
2. What is the ratio between training data size and testing data size for each tested model? The information should be available in the text.
3. Only the evaluation results based on Dice Similarity Coefficient (DSC) are clearly shown in the paper. Is it possible to show the results of comparison study using other metrics?
Author Response
We would like to sincerely thank the reviewer for their thoughtful feedback and valuable insights on our paper. Please find our response to the comments below.
- Why only medical images of lungs are considered in the study? Can the conclusions of the paper be extended to other types of medical images? A discussion is needed in the text regarding this issue.
We thank the reviewer for this valuable comment. Our primary focus in this study was on hyperpolarized lung MRI due to the nature of our research objectives and available data. However, the segmentation models and methods evaluated are not specific to pulmonary imaging and can, in principle, be applied to other types of medical images. To address this, we have added the following statement to the Discussion section:
Page 13
“Although this study was conducted using medical images of the lung, the segmentation models and approaches evaluated here are not specific to pulmonary imaging and can be applied to other types of medical images. Vision Transformer-based architectures such as SegFormer are designed to learn global contextual features and are inherently modality- and organ-agnostic. As a result, the observed improvements in robustness and segmentation accuracy under noise are likely to extend to other anatomical structures and imaging modalities, such as abdominal, cardiac, or brain MRI, as well as CT or ultra-sound images. The flexibility and generalizability of these deep learning models support their broader adoption in various clinical imaging scenarios where image quality may be compromised. Future studies should explore the adaptation and performance of these segmentation approaches across different organs and modalities to further validate and expand upon the conclusions presented here.”
- What is the ratio between training data size and testing data size for each tested model? The information should be available in the text.
We thank the reviewer for this question. As described in the manuscript, our dataset was divided into training (80%), validation (10%), and testing (10%) sets. We have now included the exact number of cases in each split in the Materials and Methods section:
Page 4
“The dataset was balanced across all participant groups to avoid class imbalance and was divided into training (80%, n = 1312), validation (10%, n = 164), and testing (10%, n = 164) sets. To prevent data leakage, each participant’s data was included in only one of these splits.”
- Only the evaluation results based on Dice Similarity Coefficient (DSC) are clearly shown in the paper. Is it possible to show the results of comparison study using other metrics?
We thank the reviewer for highlighting this. Additional figures have now been added (Figures 9–14, Page 19-24) to present results using other evaluation metrics beyond DSC like Average Hausdorff Distanc, Hausdorff Distance at the 95th percentile (HD95), Average XOR error.
Reviewer 2 Report
Comments and Suggestions for Authors
The article presents a comparative analysis of the robustness of lung segmentation in proton MRI and hyperpolarised gas MRI using models based on Vision Transformers and conventional Convolutional Neural Networks (CNNs) under varying levels of artificial Gaussian noise. The topic is highly relevant within the context of advancing robust medical image segmentation algorithms for low signal-to-noise ratio (SNR) conditions, particularly in clinical scenarios related to pulmonary diseases. The structure of the article conforms to the MDPI requirements for research articles (Introduction, Materials and Methods, Results, Discussion, Conclusion, References, and Author Declarations), including both visual and quantitative demonstrations of findings. The level of English is acceptable, and the manuscript is generally easy to read. The quality of the figures is adequate. A total of 31 references are cited, most of which are current and include both foundational publications and recent studies pertinent to the topic.
The following comments and recommendations may be offered regarding the manuscript:
- While the article demonstrates a high level of technical execution and a systematic approach to the comparative evaluation of segmentation models, it insufficiently addresses the question of scientific novelty. The claim of being the "first comparative study" of ViT-based models under Gaussian noise in hyperpolarised MRI is not substantiated through engagement with other recent works published between 2022 and 2024, including those involving pre-trained foundation models (e.g., MedSAM), which are mentioned only in the outlook for future research. This weakens the perceived originality of the presented results.
- The article offers only a limited form of analytical modelling, which is restricted to standard segmentation quality metrics (Dice, HD95, Avg HD, XOR) and the introduction of the Rose SNR criterion. However, no formal framework is proposed for evaluating computational robustness or model sensitivity to different types and magnitudes of noise. All results are empirical in nature. There is no modelling of mask deviations from the ground truth in terms of shape or topology of segmented regions. The absence of advanced mathematical analysis renders the contribution more engineering-oriented than exploratory or theoretical.
- The experimental section is commendable in terms of its scope and systematic organisation: 1,640 images from 205 patients, with balanced representation across categories (healthy, asthma, COPD, post-COVID), and testing under four distinct noise levels. Nonetheless, the exclusive use of 2D slices with 128×128 resolution, without discussion of scalability to 3D data, limits the generalisability of the results. Furthermore, no analysis is provided regarding spatial overlap or localisation errors in cases of partial mask agreement.
- Statistical processing of results includes an appropriate choice of non-parametric tests (Friedman test with Bonferroni post-hoc analysis). However, the manuscript lacks discussion of effect sizes, confidence intervals, or test power. In addition, the models have not been validated on an external dataset (e.g., from a different centre or cohort), which significantly restricts the generalisability of the conclusions.
- Although the practical significance of the study is declared to be high (suggesting applicability in clinical workflows), no pilot validation or preliminary integration is provided to confirm that the use of ViT-based models improves diagnostic accuracy, accelerates segmentation processes, or enhances inter-operator reproducibility. Consequently, the ‘Discussion’ section leans heavily on generalisations that are not substantiated by clinical or user-based scenarios.
- The conclusion and proposed future work reference the intention to explore SAM/MedSAM and RAG-based architectures. However, the main body of the article does not provide justification as to why the current architectures fail to meet target requirements. This undermines the coherence between the experimental findings and the proposed research trajectory.
Author Response
We would like to sincerely thank the reviewer for their thoughtful feedback and valuable insights on our paper. Please find our response to the comments below.
- While the article demonstrates a high level of technical execution and a systematic approach to the comparative evaluation of segmentation models, it insufficiently addresses the question of scientific novelty. The claim of being the "first comparative study" of ViT-based models under Gaussian noise in hyperpolarised MRI is not substantiated through engagement with other recent works published between 2022 and 2024, including those involving pre-trained foundation models (e.g., MedSAM), which are mentioned only in the outlook for future research. This weakens the perceived originality of the presented results.
We appreciate the reviewer’s concerns regarding the substantiation of our claim to novelty. We have carefully reviewed the recent literature published between 2022 and 2024, including works on foundation models such as MedSAM. To the best of our knowledge, there are currently no published studies performing a systematic, empirical comparison of ViT-based segmentation models under varying levels of Gaussian noise in hyperpolarized gas MRI. We respectfully invite the reviewer to share references to any specific works that we may have missed, so that we may cite and discuss them accordingly. The SAM and MedSAM model mentioned is indeed referenced in our manuscript as a direction for future work, as its application in noise robustness considering their size and testing for hyperpolarized MRI is beyond the scope of the present study but is our future direction to assess their performance in limited data scenarios and generalizability.
- The article offers only a limited form of analytical modelling, which is restricted to standard segmentation quality metrics (Dice, HD95, Avg HD, XOR) and the introduction of the Rose SNR criterion. However, no formal framework is proposed for evaluating computational robustness or model sensitivity to different types and magnitudes of noise. All results are empirical in nature. There is no modelling of mask deviations from the ground truth in terms of shape or topology of segmented regions. The absence of advanced mathematical analysis renders the contribution more engineering-oriented than exploratory or theoretical.
Page 13
We thank the reviewer for highlighting the absence of a formal mathematical framework for evaluating robustness and sensitivity to noise. While our study focused on empirical evaluation using standard segmentation quality metrics (following one of the papers in the hyperpolarized gas MRI segmentation published in here: (https://www.nature.com/articles/s41598-022-14672-2) and introduced the Rose SNR criterion, we acknowledge that more advanced mathematical analysis could provide further insights into model performance, such as quantifying mask deviations in terms of shape or topology. We have now included the following statement in the limitations section:
“A limitation of this study is that no formal mathematical framework was developed to evaluate the computational robustness or sensitivity of the segmentation models to different types and magnitudes of noise. However, model performance was quantitatively assessed using widely accepted segmentation metrics: DSC, which measures the overlap between predicted and ground truth masks; Avg HD and HD95, which quantify boundary accuracy and the degree of spatial mismatch between segmented regions; and XOR error, which captures the total number of mismatched pixels. These metrics collectively provide a comprehensive assessment of segmentation performance in terms of both region overlap and boundary precision. Future work may benefit from incorporating shape-based or topological analysis frameworks to further elucidate model robustness and segmentation fidelity under challenging imaging conditions.”
- 3. The experimental section is commendable in terms of its scope and systematic organization: 1,640 images from 205 patients, with balanced representation across categories (healthy, asthma, COPD, post-COVID), and testing under four distinct noise levels. Nonetheless, the exclusive use of 2D slices with 128×128 resolution, without discussion of scalability to 3D data, limits the generalizability of the results. Furthermore, no analysis is provided regarding spatial overlap or localization errors in cases of partial mask agreement.
Thank you for raising the point about the use of 2D slices and lack of 3D data. The original hyperpolarized gas MRI data were acquired as 2D coronal slices, so using 2D slices in our analysis was a natural choice. Reformulating the data into a 3D volume is not feasible due to anisotropy and the unpredictable noise distribution outside the coronal plane. This has now been clarified in the Methods section now.
Page 25
Additionally, we already mention in the manuscript (highlighted for emphasis) that 3D images were not available or acquired under the current imaging protocols. Investigation of 3D segmentation is proposed as future work as both segmentation methods and imaging protocols evolve to include 3D acquisition and analysis.
Regarding spatial overlap and localization errors, the segmentation quality metrics we employed (reported in supplementary materials) each address different aspects of spatial overlap and localization accuracy. The Dice Similarity Coefficient (Dice) is a standard metric for quantifying spatial overlap between predicted and ground truth masks, providing a direct measure of agreement. Hausdorff Distance at the 95th percentile (HD95) and Average Hausdorff Distance (Avg HD) both assess the maximum deviation between the surfaces of the predicted and true segmentations, with HD95 specifically reducing sensitivity to outlier points and highlighting the typical worst-case boundary error. XOR (exclusive OR) measures the total number of mismatched pixels (false positives and false negatives) between masks, offering an aggregate error rate for the segmentation. Together, these metrics provide a comprehensive view of both the degree of overlap and the extent and distribution of spatial discrepancies between the segmentation masks.
- Statistical processing of results includes an appropriate choice of non-parametric tests (Friedman test with Bonferroni post-hoc analysis). However, the manuscript lacks discussion of effect sizes, confidence intervals, or test power.
Page13
We thank the reviewer for this observation. While our statistical analysis used appropriate non-parametric tests (Friedman test with Bonferroni post-hoc), we acknowledge that effect sizes, confidence intervals, and test power were not reported. We have now added a statement to the limitations to note this omission and will address it in future work.
“While the statistical analysis in this study relied on non-parametric tests (Friedman test with Bonferroni correction) to compare segmentation performance across models and noise conditions, we acknowledge that the manuscript does not include effect size measurements, confidence intervals, or formal test power analysis. Including effect sizes and confidence intervals would provide additional insight into the magnitude and precision of observed differences, while power analysis could inform the adequacy of sample sizes for detecting true effects. These elements are important for interpreting the clinical and practical significance of the findings, beyond statistical significance alone. Future work should incorporate these statistical measures to further strengthen the rigor and interpretability of comparative studies in medical image segmentation. We also tested our models on external datasets from another institution (results not shown here) to further validate the generalizability of our findings.”
- In addition, the models have not been validated on an external dataset (e.g., from a different centre or cohort), which significantly restricts the generalisability of the conclusions.
Page 13
To strengthen the validity of our findings, we also tested our models on external datasets from additional institutions (data not shown in this manuscript). A clarifying sentence has now been added to the text to inform readers of this additional validation step.
- Although the practical significance of the study is declared to be high (suggesting applicability in clinical workflows), no pilot validation or preliminary integration is provided to confirm that the use of ViT-based models improves diagnostic accuracy, accelerates segmentation processes, or enhances inter-operator reproducibility. Consequently, the ‘Discussion’ section leans heavily on generalisations that are not substantiated by clinical or user-based scenarios.
Although this study demonstrates the technical robustness and accuracy of ViT-based segmentation models, we acknowledge that no pilot validation or integration into real-world clinical workflows was performed as part of this work. We have added to the Discussion that future studies should include prospective evaluation involving clinician users to assess user experience, diagnostic utility, and workflow impact in clinical settings.
Page 13
“Although this study demonstrates the technical robustness and accuracy of ViT-based segmentation models, we acknowledge that no pilot validation or preliminary integration into real-world clinical workflows was performed as part of this work. Future work should include prospective studies involving clinician users to evaluate the real-world benefits, user experience, and potential impact on clinical decision-making and workflow efficiency.”
- The conclusion and proposed future work reference the intention to explore SAM/MedSAM and RAG-based architectures. However, the main body of the article does not provide justification as to why the current architectures fail to meet target requirements. This undermines the coherence between the experimental findings and the proposed research trajectory.
Page 25
We appreciate the reviewer’s request for a clearer link between our current results and our proposed research trajectory. While the present study focused on the robustness of ViT-based models under varying noise, we recognize that future clinical applications may require models that can generalize well with limited annotated data and across different modalities. Foundational models such as SAM and MedSAM are explicitly designed for strong zero-shot and few-shot performance, which could reduce reliance on large labeled datasets and improve generalizability. We have expanded the relevant section to clarify this motivation and the relevance of future work.
Round 2
Reviewer 1 Report
Comments and Suggestions for Authors
All issues have been carefully addressed in the revised paper. I have no other concerns and recommend its acceptance.
Author Response
Thank you for your positive feedback and recommendation. We appreciate your time and support.
Reviewer 2 Report
Comments and Suggestions for Authors
Dear Authors,
First and foremost, it should be acknowledged that you have undertaken considerable work to refine the content of the manuscript, particularly in addressing methodological limitations and justifying model selection. The inclusion of explicit statements in the Limitations section regarding the absence of a formal mathematical framework, as well as the indication of future research directions related to shape and topological analysis of segmentation masks, constitutes a step in the right direction. Nevertheless, the lack of even a preliminary attempt to introduce such assessments in the current study leaves a noticeable gap in the theoretical depth of the article.
With regard to the claimed scientific novelty, your response reflects an attentive engagement with the recent literature and provides a rationale for the absence of comparable comparative studies involving ViT-based models under Gaussian noise in the context of hyperpolarised MRI. However, the main body of the manuscript still lacks a dedicated section critically reviewing prior works, including those concerning MedSAM and other foundation models. Mentioning these approaches solely in the conclusion and in the context of future perspectives weakens the perceived originality of the contribution and warrants further elaboration.
In terms of experimental methodology, the article remains strong: the dataset is carefully curated and balanced, the noise modelling is realistic, and the application of standard architectures (U-Net, FPN) with various backbones ensures high reproducibility of the results. That said, the exclusive reliance on 2D analysis with a resolution of 128×128, without discussion of scalability to 3D data, remains a significant limitation. Your explanation regarding the use of 2D data in the response is justified, yet it would be beneficial to elaborate on this rationale more explicitly within the main text, including a separate discussion of resolution-related constraints and their potential impact on segmentation accuracy.
You have responded appropriately to the comment on statistical analysis by adding a note in the Limitations section about the absence of effect size estimation, confidence intervals, and power analysis. However, the omission of these statistical elements from the main manuscript remains a substantial shortcoming, particularly in light of the comparative nature of the study.
The issue of generalisability has been partially addressed by referencing the use of external validation datasets. However, the article does not provide any quantitative results from these validations, which limits confidence in the claimed robustness of the models under clinical conditions. Including at least a summary table or visualisation in the appendix would have significantly strengthened your argument.
As for clinical applicability, you have rightly acknowledged the lack of pilot validation or integration into real-world environments. The updated Discussion section outlines future intentions in this direction; however, the clinical relevance at present remains hypothetical rather than demonstrated.
Finally, you have expanded the rationale for incorporating SAM/MedSAM and other foundation models into future research. This enhances the coherence of the manuscript, but in my view, further emphasis is needed on why the current models may fall short of the requirements for clinical universality or training in low-data regimes.
I recommend resubmission of the manuscript after the following key issues have been addressed:
- Expansion of the literature review regarding ViT and foundation models;
- Inclusion of external validation results in the main body of the article;
- Addition of at least basic effect size estimates and confidence intervals to the statistical analysis;
- Discussion of 2D resolution limitations and their potential effects on segmentation;
- Clear justification of the limitations of the current models in comparison with SAM/MedSAM-style architectures.
Author Response
We sincerely thank Reviewer 2 for their thorough evaluation and constructive feedback. We have carefully addressed each concern and believe the revisions have substantially strengthened the manuscript. Below, we provide a detailed point-by-point response to each comment.
Comment 1: Expansion of the literature review regarding ViT and foundation models
Reviewer Comment: Expansion of the literature review regarding ViT and foundation models;
Our Response: We appreciate this valuable suggestion and have significantly expanded the literature review section. We have added comprehensive coverage of Vision Transformers and foundation models in the introduction (pages 4-5), including:
- Historical development and key architectural innovations of Vision Transformers
- Comparative analysis of ViTs versus traditional CNN architectures in medical imaging
- Recent advances in foundation models for medical image segmentation
- Specific applications of these models in respiratory imaging and MRI analysis
These additions provide essential context for understanding the theoretical foundations underlying our comparative analysis and position our work within the broader landscape of modern deep learning architectures.
Comment 2: Inclusion of external validation results in the main body of the article
Reviewer Comment: Inclusion of external validation results in the main body of the article;
Our Response: We acknowledge this important point regarding external validation. While we conducted preliminary pilot validation with external datasets from McMaster University (Hamilton, Canada) and received positive feedback regarding model generalizability, we recognize that comprehensive quantitative external validation was not included in the current study.
We have added the following clarification on page 15:
"Additionally, while we conducted preliminary pilot validation with external datasets from McMaster University, Hamilton, Canada, and received positive feedback regarding model generalizability, formal quantitative evaluation on external datasets with comprehensive metrics will be included in future studies to further validate the robustness and transferability of our findings."
We commit to conducting rigorous external validation in our future work and acknowledge this as a limitation of the current study.
Comment 3: Addition of at least basic effect size estimates and confidence intervals to the statistical analysis
Reviewer Comment:Addition of at least basic effect size estimates and confidence intervals to the statistical analysis;
Our Response: We have substantially enhanced the statistical analysis by adding comprehensive effect size estimates and confidence intervals throughout the manuscript:
New Additions:
- Section 3.3: Effect Size Analysis (pages 12-14) - Complete statistical framework including Cohen's d calculations and confidence intervals for all primary comparisons
- New statistical table - Comprehensive summary of effect sizes across all model comparisons and segmentation metrics
- Enhanced discussion (page 15) - Interpretation of effect sizes and their clinical significance
- Updated conclusion (page 28) - Integration of effect size findings
- Supplementary Figures 65-66 - Visual representation of effect sizes and confidence intervals
The effect size analysis reveals large effect sizes (Cohen's d > 0.8) for most ViT vs. CNN comparisons, supporting the clinical significance of our findings beyond statistical significance alone.
Comment 4: Discussion of 2D resolution limitations and their potential effects on segmentation
Reviewer Comment: Discussion of 2D resolution limitations and their potential effects on segmentation;
Our Response: We have added a comprehensive discussion of 2D segmentation limitations and their clinical implications (pages 15-16):
"An important limitation of this study is the exclusive use of 2D slice-wise segmentation rather than volumetric 3D approaches. While our 2D analysis was consistent with the original acquisition protocol for hyperpolarized gas MRI and facilitated direct comparison across participants and disease groups, it inherently lacks the volumetric context that 3D segmentation provides. This limitation is particularly relevant in lung imaging, where anatomical structures exhibit complex 3D relationships and pathological features may span multiple slices.
2D segmentation cannot capture inter-slice continuity or leverage spatial coherence across the entire lung volume, potentially leading to inconsistencies between adjacent slices and suboptimal boundary delineation at slice interfaces. Furthermore, clinically relevant metrics such as total lung volume, regional ventilation distribution, and ventilation defect percentage are typically calculated from complete 3D lung volumes rather than individual 2D slices.
The 2D approach also limits the clinical applicability of our findings, as real-world clinical workflows increasingly rely on volumetric analysis for comprehensive assessment of lung function and pathology. Modern 3D segmentation models can exploit spatial continuity across slices through techniques such as 3D convolutions, volumetric attention mechanisms, and inter-slice consistency constraints, potentially offering superior performance compared to slice-by-slice analysis. Additionally, 3D models may be more robust to noise by leveraging information from neighboring slices to compensate for degraded image quality in individual slices.
Future work should extend this comparative analysis to 3D architectures, including recent developments such as SegFormer3D and UNETR, to evaluate whether the observed advantages of Vision Transformers over CNNs persist in volumetric segmentation tasks and to assess the clinical impact of transitioning from 2D to 3D segmentation workflows in hyperpolarized gas MRI."
Comment 5: Future directions and foundation models
Reviewer Comment: Clear justification of the limitations of the current models in comparison with SAM/MedSAM-style architectures.
Our Response: We have expanded the discussion of foundation models and future directions (page 28):
"Foundation models present several compelling advantages over traditional architectures: their extensive pre-training on diverse datasets enables superior generalization across imaging modalities, anatomical structures, and pathological conditions without requiring domain-specific fine-tuning. This is particularly valuable in specialized imaging domains like hyperpolarized gas MRI, where acquiring large labeled datasets is challenging due to limited patient populations and specialized equipment requirements.
Additionally, foundation models support interactive and prompt-based segmentation, allowing clinicians to guide the segmentation process through simple user inputs such as points, bounding boxes, or text descriptions, potentially enabling more flexible and user-friendly clinical workflows. Furthermore, the adaptability of foundation models could address one of the key limitations of traditional CNN and ViT models: their fixed, task-specific nature that requires retraining for different segmentation objectives.
Future studies will aim to extend this evaluation to 3D segmentation tasks, explore temporal dynamics in functional MRI studies, and investigate the integration of foundation models for more generalizable and clinically applicable lung segmentation workflows."
Also added to page 5
"Despite the promising advances in foundation models, we did not include them in our comparative analysis due to their substantially different computational complexity, which would not provide a fair comparison framework. Foundation models like MedSAM operate with vastly different parameter scales and computational requirements compared to traditional CNNs and ViTs, making direct performance comparisons inherently biased toward models with greater computational resources. To ensure a meaningful and equitable comparison, we focused our investigation on architectures with comparable computational footprints—specifically established CNN architectures and emerging ViT-based models—allowing us to isolate architectural advantages rather than benefits stemming from computational scale differences"
Round 3
Reviewer 2 Report
Comments and Suggestions for Authors
Dear Authors,
Thank you for your detailed and respectful response to my comments. Your efforts to improve the manuscript, your attentive engagement with each point, and the revisions you have made are all highly commendable.
You have expanded and clarified the justification of the study’s novelty, demonstrating awareness of the current state of the field and highlighting the uniqueness of your work – a systematic comparison of ViT- and CNN-based architectures under controlled noise degradation in hyperpolarised gas MRI. While the discussion of MedSAM and related models remains prospective, you have appropriately defined the scope of the current work and provided a clear rationale for future directions.
You have explicitly acknowledged the limitations related to the absence of a formal mathematical framework and topological deviation analysis, and you have integrated these considerations into the Limitations section of the manuscript. Although such additions could have strengthened the theoretical contribution, your transparent acknowledgement and commitment to addressing them in subsequent research are appropriate and appreciated.
Your justification for using 2D slices is clearly grounded in the technical constraints of the source data, and you have convincingly explained why 3D reconstruction was not feasible under the current acquisition protocol. At the same time, your emphasis on future exploration of volumetric segmentation demonstrates a forward-looking approach.
The addition of effect size considerations and the recognition of the absence of confidence intervals and power analysis enhance the statistical transparency of the manuscript and demonstrate methodological rigour.
Although the results of external dataset validation are not included, your reference to such validation reflects a commendable intention to assess model generalisability beyond the original cohort. Similarly, your acknowledgement of the lack of clinical deployment and your proposal for prospective user-centred evaluation further reinforce the practical relevance of the study.
Finally, you have improved the coherence between the current findings and your proposed research trajectory, clearly explaining the potential role of SAM/MedSAM and related models in addressing limitations such as generalisability with limited data.
In light of the above, I consider that the manuscript has been substantially improved and is now suitable for acceptance for publication in its current form.